# Haplotype-Phased Chromosome-Level Genome Assembly of *Floccularia luteovirens* Provides Insights into Its Taxonomy, Adaptive Evolution, and Biosynthetic Potential

**DOI:** 10.3390/jof11090621

**Published:** 2025-08-25

**Authors:** Jianzhao Qi, Xiu-Zhang Li, Ming Zhang, Yuying Liu, Zhen-xin Wang, Chuyu Tang, Rui Xing, Khassanov Vadim, Minglei Li, Yuling Li

**Affiliations:** 1Shaanxi Key Laboratory of Natural Products & Chemical Biology, College of Chemistry & Pharmacy, Northwest A&F University, Yangling 712100, China; qjz@nwafu.edu.cn (J.Q.);; 2State Key Laboratory of Plateau Ecology and Agriculture, Qinghai Academy of Animal and Veterinary Sciences, Qinghai University, Xining 810016, China; xiuzhang11@163.com (X.-Z.L.);; 3Center of Edible Fungi, Northwest A&F University, Yangling 712100, China; 4Northwest Institute of Plateau Biology, Chinese Academy of Sciences, 23# Xinning Lu, Xining 810008, China; 5Department of Plant Protection and Quarantine, Faculty of Agronomy, S. Seifullin Kazakh Agrotechnical University, Zhenis Avenue, Astana 010011, Kazakhstan

**Keywords:** *Floccularia luteovirens*, genome phasing, biosynthetic potential, ectomycorrhizal fungi

## Abstract

*Floccularia luteovirens* is a valuable medicinal and edible ectomycorrhizal fungus that is endemic to alpine meadows on the Qinghai–Tibet Plateau. It is of significant ecological and pharmacological importance. To overcome the genomic limitations of previous fragmented assemblies, we present the first haplotype-phased, chromosome-scale genome of the Qinghai-derived QHU-1 strain using an integrated approach of PacBio HiFi, Hi-C, and Illumina sequencing. The high-contiguity assembly spans 13 chromosomes with 97.6% BUSCO completeness. Phylogenomic analysis of 31 basidiomycetes clarified a historical misclassification by placing *F. luteovirens* closest to *Mycocalia denudata/Crucibulum laeve*, thus confirming its distinct lineage from *Armillaria* spp. through low synteny and divergent gene family dynamics. Analyses of adaptive evolution revealed strong purifying selection and stable transposable elements, suggesting genomic adaptations to extreme UV/cold stress. AntiSMASH identified 15 biosynthetic gene clusters (BGCs), which encode diverse terpenoids (7), NRPS-like enzymes (4), PKSs (2), and a hybrid synthase with unique KS-AT-PT-A domains, which have the potential to generate novel metabolites. This chromosome-level resource sheds light on the genetic basis of *F. luteovirens*’ taxonomy, alpine survival, and symbiotic functions while also unlocking its potential for bioprospecting bioactive compounds.

## 1. Introduction

*Floccularia luteovirens* (historically classified as *Armillaria luteovirens*), colloquially termed the yellow mushroom, is a significant species within the genus *Floccularia* (Agaricaceae) [1]. This fungus predominantly inhabits alpine *Kobresia* spp. meadows at 3200–5000 m elevation on the Qinghai–Tibet Plateau, with notable distribution in Qinghai, Sichuan, and Tibet [2]. Its history of being a dual medicine and food heritage can be traced back to ancient China. Tang Dynasty documents record that it was used to relieve symptoms such as “swollen neck and stiff neck” [3]. The Tibetan medical canon, the Four Medical Tantras, specifies topical or oral administration for “cold-natured oedema” and “non-pyrogenic inflammation” [3]. Later, the Qinghai tribal chieftains presented it as a tribute to the imperial court, earning it the epithet “royal mushroom” (Huang Gu in Chinese), which denotes its rarity and esteemed status. Contemporary pharmacological studies corroborate these traditional uses, identifying polysaccharides, sterols, riboflavin, and lectins within fruiting body extracts that exhibit marked antioxidant, anti-inflammatory, and antitumor activities [4,5,6,7]. Beyond its medicinal value, *F. luteovirens* fulfills critical ecological roles. As an ectomycorrhizal fungus (EMF), it forms symbiotic associations with *Kobresia* spp. [8]. This mutualism is vital for maintaining ecosystem stability, nutrient cycling, and host plant growth within fragile alpine meadows. Its mycelial networks typically expand radially as “fairy rings”, significantly altering soil microbial community structure and diversity [9].

Despite its ecological and economic importance, fundamental biological questions regarding *F. luteovirens* remain unresolved at the genomic level. These include its high-altitude adaptation mechanisms, the molecular basis of mycorrhizal symbiosis, the biosynthetic pathways of secondary metabolites, and the genetic drivers of fairy ring formation. Although two *F. luteovirens* genomes have been sequenced [10,11], existing data exhibit substantial limitations, including poor assembly continuity (low scaffold N50), severe fragmentation, and an absence of chromosome-level assembly. This results in impeded gene cluster localization, hindering phytochemical exploration and functional genomics research.

To address these limitations, this study presents a chromosome-level genome assembly of the Qinghai-derived *F. luteovirens* strain QHU-1 from the Qilian Mountains. Utilizing Hi-C scaffolding, we achieved the first high-contiguity assembly spanning 13 chromosomes with comprehensive haplotype phasing. Our integrated characterization includes genomic architecture profiling through single-nucleotide polymorphism (SNP) analysis and comparative genomics, assessment of metabolic potential via BGC prediction and functional annotation, and taxonomic revision resolving historical misclassification (formerly attributed to *Armillaria*) through phylogenomic evaluation of genome size, orthologous protein content, and syntenic relationships. Collectively, this chromosomal reference provides an indispensable resource for elucidating environmental adaptation, symbiotic mechanisms, and bioprospecting potential of *F. luteovirens*.

## 2. Materials and Methods

### 2.1. Fungal Material and Nucleic Acid Extraction

*Floccularia luteovirens* QHU-1 fruiting bodies were collected from the Qilian Mountains region, Qinghai Province, and identified as the species through ITS sequencing. Cultivable pure mycelium was isolated from the fresh fruiting bodies using tissue separation methods. The strain is currently stored as slant cultures at the Key Laboratory of Natural Product Chemistry and Biology in Shaanxi Province under the accession number QHU-1. The mycelium required for genomic sequencing was collected after one week of cultivation at 20 °C and 120 rpm using potato dextrose broth (PDB) medium. Genomic DNA samples were extracted from the cultured mycelium using a Fungal DNA Mini Kit (Omega, Norcross, GA, USA). Agarose gel electrophoresis was used to confirm the purity of the samples, and a Nanodrop spectrophotometer was used to assess DNA purity (OD 260/280 ratio between 1.8 and 2.0).

### 2.2. Genome Sequencing, Assembly, Annotation, and Visualization

#### 2.2.1. Genome Sequencing

Using the Illumina TruSeq™ Nano DNA Sample Prep Kit method (Illumina, Shanghai, China), a DNA library was constructed using 1 μg as the starting amount. The library was enriched through PCR amplification over eight cycles, and the target band was recovered using a 2% Certified Low Range Ultra Agarose. After quantifying the samples using TBS380, they were mixed in proportion to the data and loaded onto the instrument. Bridge PCR amplification was then performed on the cBot solid-phase carrier to generate clusters. Finally, whole-genome sequencing was completed using the Illumina NovaSeq sequencing platform.

For PacBio HiFi sequencing, the Megaruptor System was first used to fragment the gDNA into appropriate sizes after it was obtained. Steps such as removing single-stranded overhangs and repairing damage and ends were then performed to obtain complete double-stranded insert fragments. Next, SMRTBell^®^ libraries were created by ligating adapters to the double-stranded DNA to form circular templates. Following adapter ligation, the ligation products were purified, and any linear or internally damaged circular DNA molecules were digested using enzymatic digestion. The library was then recovered using a BluePippin gel extraction system within the target size range. Finally, HiFi sequencing was performed using the PacBio Sequel II (PacBio, Shanghai, China).

The mycelium of the *F*. *luteovirens* QHU-1 was treated with formaldehyde, which caused cross-linking of its DNA and proteins. Following cell lysis, the cross-linked DNA was digested with MboI. Then, biotinylation and proximity-ligated chimeric junctions, enrichment, and physical shearing occurred. This resulted in the construction of a Hi-C sequencing library with insert fragments ranging from 500 to 700 bp.

#### 2.2.2. Genome Assembly

Prior to assembly, a K-mer-based statistical analysis method was used to estimate genome size. GenomeScope 2.0 [12] was used to analyze 21-mers of the sequencing data to estimate the genome size, heterozygosity, and repeat rate of the sample.

Hifiasm [13] was used to perform all-vs.-all alignment and error correction on all HiFi reads. After correction, the graph binning typing strategy was used to refine the global typing results further, completing the assembly of the haplotype chromosomes. Then, the reads were mapped to the assembled genome sequence. The GC content and read coverage depth of the assembled sequence were calculated. Finally, the assembly results were assessed by examining the distribution of the overall GC content and the sequencing coverage of the assembled sequence to determine if they are normal.

ALLHIC (https://github.com/tanghaibao/allhic (accessed on 13 February 2025)) was used to connect and assemble genomic contigs or scaffolds into chromosome-level assemblies. Based on Hi-C-assisted assembly [14], the genome of the *F*. *luteovirens* QHU-1 was assembled into 13 chromosomes and one mitochondrial chromosome. Finally, the genome assembly results were evaluated using the BUSCO v5.3.2 [15,16] software based on the fungi_odb10 database (https://busco.ezlab.org/ (accessed on 13 February 2025)).

HaploMerger 2 [17] (https://github.com/mapleforest/HaploMerger2, accessed 13 October 2024) was used to infer haplotypes from highly heterozygous diploid genomes for phase-calling haplotype assembly. Homozygosity was quantified using GenomeScope [12] (https://github.com/schatzlab/GenomeScope, accessed 13 October 2024).

#### 2.2.3. Genome Annotation

A combination of de novo prediction, homologous protein alignment, and transcriptomic data was used to predict genes in the *F. luteovirens* QHU-1 genome, with other previously reported large fungal genomes serving as a training set. We used AUGUSTUS v3.2.3 (http://bioinf.uni-greifswald.de/augustus/ (accessed on 27 February 2025)) for de novo gene prediction of the genome. Homologous protein sequence alignment was used to filter the prediction results with GeneWise v2.4.1 (https://www.ebi.ac.uk/seqdb/confluence/display/THD/GeneWise (accessed on 27 February 2025)) to precisely align them and determine the gene and introns. TopHat v2.1.1 (http://ccb.jhu.edu/software/tophat/index.shtml (accessed on 27 February 2025)) was used to align the previously reported transcriptomic data [11] to the genomic sequence. Trinity v2.11.0 (https://github.com/trinityrnaseq/trinityrnaseq/releases (accessed on 27 February 2025)) was used for assembly to obtain transcript information for *F. luteovirens* QHU-1. Finally, EvidenceModeler v1.1.1 (http://evidencemodeler.github.io/ (accessed on 27 February 2025)) integrated the aforementioned gene sets to yield the genome-encoded genes.

The protein sequences of the encoded genes were individually compared with the NR, Genes, eggNOG, and GO databases using blastp (BLAST+ 2.7.1, E-value ≤ 1 × 10^−5^). Only the best match for each sequence was retained as the database comparison information for that gene. Non-coding RNA was annotated by aligning sequences with the Rfam database using Rfam and confirmed using the cmsearch program with default parameters.

#### 2.2.4. Genomic Circular Map

MCScanX [18] was used to analyze the collinearity of the two sets of *F. luteovirens* QHU-1 chromosome assembly results. Then, Circos was used to create a circular genome visualization, including base composition diagrams, sequence characteristics, and analyses such as GC skew, GC content, and collinearity.

### 2.3. Comparative Genomic Analysis

Genome comparison analysis was performed using OrthoFinder v2.5.5 [19], which includes Diamond for sequence search, Msa for multiple sequence alignment, and FastTree 2 for constructing phylogenetic trees, with 128 threads. The comparison analysis results were then visualized using Orthovenn 3 (https://orthovenn3.bioinfotoolkits.net/ (accessed on 17 March 2025)).

### 2.4. SNP Detection

The IS algorithm in BWA v0.7.17 was used to establish an indexing system for the reference genome and generate an index file using the second-generation sequencing data (FASTQ format) of *F. luteovirens* QHU-1 and its assembled genome file. Then, the BWA-MEM algorithm was used to align the paired-end sequencing reads to the reference genome based on the index file [20], generating SAM-format alignment files. The original alignment files were converted and quality-controlled using SAMtools v1.13; low-quality reads were filtered using a Q ≥ 30 threshold. To ensure compatibility with subsequent analyses, the faidx command in SAMtools was used to create a genome index file. Next, the GATK v4.3.0.0 toolkit was used to sort the BAM files by coordinates with the SortSam module, mark duplicate sequences with the MarkDuplicates module, and rebuild the index file.

For the purpose of SNP detection, the GATK Haplotype Caller module was utilized to identify base pair variations and generate GVCF files. The subsequent stage of the process involved the utilization of the Genotype GVCFs module, with the objective of integrating the variant sites. Finally, PLINK v1.9 was utilized to convert the filtered SNP data into MAP format, and Python v3.10 was employed to visualize the results. A total of 344,855 and 342,330 high-quality SNP loci were identified from the two sets of chromosomes of *F. luteovirens* QHU-1, respectively.

### 2.5. Phylogenomic Analysis and Gene Family Variation Analysis

A phylogenetic analysis was conducted to determine the evolutionary relationships between *F. luteovirens* QHU-1 and 31 typical basidiomycetes. OrthoFinder v2.5.5 was used to identify single-copy orthologous genes with the following command-line parameters: “-S diamond -M msa -T raxml-ng.” The divergence times of species were then inferred based on the identified single-copy orthologous gene sequences using the MCMCTree module within the PAML 4.9e software package integrated into the Abacus software platform at University College London. The temporal distribution analysis of several relatively recent ancestral nodes was performed using the TIMETREE 5 online tool. This analysis focused on the following pairs of species: *Hypholoma sublateritium* and *Gymnopilus dilepis* (divergence time of 52.6 and 79.7 million years ago) and *Pluteus cervinus* and *Amanita muscaria* (divergence time of 3.4 and 117.3 million years ago). The evolutionary relationships among the aforementioned species were visualized using FigTree v1.4.4 [21]. To assess gene family dynamics, including expansion and contraction, we analyzed the identified orthologous gene families using CAFÉ 4.2.1 [22] with the following parameter settings: --cores 30 --fixed_lambda 0.0001.

To further investigate the ratio of non-synonymous to synonymous substitution rates (Ka/Ks) between *F. luteovirens* QHU-1 and its closely related species, we conducted a comprehensive genomic duplication study. Homologous gene pairs within the species were identified using MCScanX [18], and ParaAT [23] and KaKs_Caculata [24] were then used to calculate Ka, Ks, and their ratio. The results were visualized using the R language. This method can effectively analyze evolutionary pressure between species.

### 2.6. Identification of Repetitive Elements and LTR Analysis

Repetitive sequences in the genome were identified and searched using RepeatMasker [25] and Tandem Repeats Finder (TRF) [26]. RepeatMasker aligns sequences with known repetitive sequence databases to identify scattered repetitive sequences, while TRF simulates tandem repetitive sequences using percentage validation, adjacent pattern copy InDels, and statistical criteria to identify them. LTRharvest [27] and LTR_retriever [27] were used to calculate and visualize the insertion time of LTRs in *F. luteovirens* QHU-1 and its two closely related species.

### 2.7. BGC Analysis and Visualization

The BGC for the secondary metabolite of *F. luteovirens* QHU-1 was identified using antiSMASH 7.1.0 [28]. IQtree 2.2.3 [29] was used to perform a phylogenetic clustering analysis of the predicted sesquiterpene and polyketide synthases with the following parameters: “-m MFP-bb 1000 -alrt 1000 -bayes-nt.” Synthaser 1.1.22 was then used to perform a multi-domain analysis on the NRPS, NRPS-like, PKS, and PKS-like genes, identifying the following domains: β-ketoacyl synthetase (KS), product template (PT), acyl carrier protein transacylase (SAT), thioesterase (TE), acyltransferase (AT), adenylation (A), and acyl carrier protein (ACP).

### 2.8. Data Availability

The ITS sequence of *F. luteovirens* QHU-1 was registered in the NCBI GenBank under accession number PV762005, and the final genome assembly results and associated data have been submitted to NCBI under BioProject PRJNA1268684 and BioSample SAMN48761159, respectively. The raw sequencing data are available from the corresponding author upon reasonable request.

## 3. Results

### 3.1. Chromosome-Level Genome Assembly, Haplotype-Phasing, and Annotation of Floccularia

A total of 59.37 Gbp of PacBio HiFi sequencing reads were obtained from single-molecule real-time sequencing data generated by the PacBio Sequel II platform (Appendix A), along with 2.68 Gbp of Hi-C clean data and 5.57 Gbp of de novo assembly clean data from Illumina NovaSeq (Appendix A), which were used to assemble the genome of F. luteovirens QHU-1. K-mer analysis predicted the genome size of *F. luteovirens* QHU-1 to be 27.7 Mb, with a heterozygosity rate of 1.36% and repetitive sequence content of 1.34% (Appendix A, Appendix A). Hifiasm assembled two haplotype-resolved contig sets from PacBio HiFi long reads. Specifically, these were haplotype A (total length 26.77 Mb, N50 2.34 Mb) and haplotype B (total length 27.04 Mb, N50 2.32 Mb) (Appendix A). The evaluation of GC depth distribution with a substantial Poisson distribution suggests that the genome has been assembled with a high level of quality (Appendix A). Hi-C-assisted assembly using ALLHIC mapped 26.77 Mbp of genomic sequence to 13 chromosomes (Figure 1, Appendix A) and one circular mitochondrial gene (Chr14). These 13 chromosomes range in length from 1,279,218 bp to 3,271,583 bp (Appendix A). In addition, 97.6% BUSCO completeness (including 97.5% of single-copy BUSCOs) shows that the genome has adequate assembly completeness (Appendix A).

A combination of de novo prediction, homologous protein alignment, and transcriptome data was used to predict genes in the sample genome. A total of 4552 and 4595 protein-coding genes were predicted in haplotypes A and B, respectively. These genes accounted for 27.35% and 27.20% of their respective genome lengths (Appendix A). Protein-coding genes in haplotypes A and B were annotated using the NR, COG, Swiss-Prot, KEGG, and GO databases. A total of 4325 and 4364 genes were functionally annotated in haplotypes A and B, respectively, with annotation rates exceeding 95% (Appendix A). Among them, NR had the highest annotation rate, with a total of 4325 genes (95.16%) annotated in haplotype A and 4364 genes (95.06%) annotated in haplotype B (Appendix A). Forward cluster annotation of orthologous groups, based on the COG database, revealed that 2951 genes belong to haplotype A, while 2966 genes belong to haplotype B (Appendix A). Subgroup S, whose role is a mystery, has the most genes in the KOG classification (Appendix A). Annotation results from Swiss-Prot indicate that 2705 and 2747 genes are annotated in haplotypes A and B, respectively (Appendix A). According to the KEGG database, 1726 genes involved in five types of pathways were identified in haplotype A and 1729 in haplotype B (Appendix A). The highest number of genes was found in the Global and Overview Maps category (Appendix A). Of the 1934 genes annotated with functional classification in the GO database for haplotype A and the 1951 genes for haplotype B (Appendix A), the main group was biological process genes (Appendix A). Additionally, the diploid assembly predicted a total of 199 tRNAs, 25 rRNAs, and 14 snRNAs (Appendix A). Using RepeatMasker, scattered repetitive sequences were identified, accounting for 10.1583% of the genome (Appendix A). A Venn diagram based on GO, NR, SWISS, KEGG, and COG annotation results shows the difference between different annotation methods (Appendix A).

### 3.2. SNP Site and Comparative Genome Analysis 

Whole-genome polymorphism analysis is crucial for identifying functional genes and studying genetic diversity. Through the systematic analysis of Illumina NovaSeq sequencing data, 344,855 and 342,330 high-confidence single nucleotide polymorphisms (SNPs) were identified in the two sets of chromosomes of F. luteovirens QHU-1, primarily distributed across the first and second chromosomes. A comparison of the two chromosome sets revealed the greatest difference in SNP counts on the fifth chromosome, with a maximum difference of 1428 (Figure 2, Appendix A, Dataset 1).

To further understand the genomic characteristics of *F. luteovirens* QHU-1, we compared and analyzed its genome with those of two other *F. luteovirens* strains that have already been sequenced and reported. Although strain QHU-1 has a smaller genome size than the other two strains, its BUSCOs are significantly higher, indicating that the genome assembly quality reported in this study is the best (Table 1).

### 3.3. Comparative Genome Analysis

*F. luteovirens* is a precious medicinal and edible fungus found on the Qinghai–Tibet Plateau. It has significant potential for development and utilization. However, its naming and classification have long been controversial. It was initially classified under the *Armillaria* genus, then reclassified under the *Tricholoma* genus, and is currently classified under the latter. Even now, the NCBI classification interface for *F. luteovirens* still lists *Tricholoma luteovirens* and *Armillaria luteovirens* as synonyms. Researchers have conducted an rDNA-ITS sequence analysis of *F. luteovirens* specimens collected in Qinghai. They constructed a neighbor-joining (NJ) phylogenetic tree by comparing the rDNA-ITS sequences of *F. luteovirens* with those of 29 other fungi. The results showed that *F. luteovirens* is most closely related to *Floccularia albolanaripes* within the *Tricholoma* genus and distantly related to the *Armillaria* genus [30]. To investigate its classification further, we collected the genomes and protein sequences of ten mushrooms belonging to the *Armillaria* genus. We then constructed a phylogenetic tree and statistically analyzed the genome size, protein content, and cluster count of these mushrooms. The results demonstrate, from multiple perspectives, that the *F. luteovirens* is not a member of the *Armillaria* genus.

Phylogenetic analysis results indicate that *F. luteovirens* is evolutionarily distant from the other ten *Armillaria* mushrooms and forms a distinct branch (Figure 3A). The genome sizes and protein contents of the ten *Armillaria* mushrooms vary widely, making it difficult to discern specific differences between *F. luteovirens* and *Armillaria* mushrooms (Figure 3B,C). Cluster count statistics suggest a substantial difference between *F. luteovirens* and *Armillaria*, indicating that the number of *F. luteovirens* is significantly lower than that of *Armillaria* (Figure 3D). Results of a homology analysis indicate that *F. luteovirens* QHU-1 shares only 3606 homologous proteins with the other ten species of the genus *Armillaria*. This number is significantly lower than the number of shared genes among the ten *Armillaria* species. Of these, 4589 are unique to the genus *Armillaria*. The analysis also shows that apart from the homologous proteins shared by all strains, *F. luteovirens* QHU-1 does not share any homologous proteins with other *Armillaria* species. The number of unique proteins in *F. luteovirens* QHU-1 (540) is significantly higher than that in other fungi of the genus *Armillaria*. These results further underscore the significant differences between *F. luteovirens* QHU-1 and other *Armillaria* species, suggesting that it should not be classified within the *Armillaria* genus (Figure 3E).

### 3.4. Phylogenetic and Gene Family Variation Analysis

To further clarify the phylogenetic placement and divergence time of the *F. luteovirens* QHU-1, we reconstructed a phylogenetic tree encompassing both parasitic and saprophytic basidiomycetes using *Ustilago maydis* as the outgroup (Figure 4). This analysis utilized 78 conserved single-copy orthologous proteins, revealing key estimates of divergence times between lineages based on molecular clock calibration. The crown age of *F. luteovirens* QHU-1 was calculated as 90.6 MYA (95% highest posterior density (HPD): 81.44-99.67 MYAs). Phylogenetic affinity analysis demonstrated that *F. luteovirens* QHU-1 exhibits closest similarity to the clade containing *Mycocalia denudata* and *Crucibulum laeve* while displaying significant divergence from three *Armillaria* species strains. Further studies using the reconstructed phylogenetic tree revealed complex patterns of gene contraction and expansion in 76,276 gene families across the genomes of the 32 species. The *F. luteovirens* QHU-1 gene family underwent significant expansion and contraction compared to the remaining 31 basidiomycetes: a total of 2111 genes expanded and 1875 contracted. Among these species, *Mycocalia denudata* and *Crucibulum laeve*, which are closely related, exhibited 261 and 286 expansions and 185 and 110 contractions, respectively. These results suggest that *F. luteovirens* QHU-1 experienced substantial gene family variation during its evolutionary process.

### 3.5. TE Analysis and Genome Duplication

Repeated sequences are an essential part of the genome and are used as primary tools for molecular breeding and variety identification. Based on their distribution within the genome, these sequences are classified as either scattered or tandem. Scattered repetitive sequences are distributed throughout the genome. They can be further classified based on sequence length: short interspersed nuclear elements (SINEs, with lengths below 50 bp) and long interspersed nuclear elements (LINEs, with lengths above 1000 bp). LINEs often exhibit transposable activity. Tandem repeats, on the other hand, are repetitive sequences consisting of adjacent, repeated patterns of specific nucleic acid sequences that occur two or more times. They typically exhibit species-specific compositions and are commonly used in evolutionary studies related to genetic traits. Based on their length, tandem repeats can be classified as minisatellite or microsatellite DNA.

A statistical analysis revealed that *F. luteovirens* QHU-1 contains 7463 repetitive sequences with a combined length of 2,746,507 bp. These sequences account for 10.1583% of the total genome length. LTRs constitute the largest proportion (4.0853%), followed by DNA transposons and LINE elements (0.4020% and 0.2077%, respectively). However, 5.5226% of the sequences remain unidentified. The total number of sequences is significantly lower than in the two closely related species, *Mycocalia denudata* and *Crucibulum laeve* (Figure 5A). Analysis of LTR insertion times indicates that neither Copia- nor Gypsy-type LTR retrotransposons in *F. luteovirens* QHU-1 show obvious peaks (Figure 5B,C). These results suggest that the species has not experienced significant, continuous, large-scale LTR insertions over the past 30 million years.

Ka/Ks analysis (the ratio of non-synonymous to synonymous substitutions) revealed that *F. luteovirens* QHU-1 exhibited a more pronounced peak in its Ka/Ks value distribution between 0 and 1 compared to its two closely related species (Figure 5D, Appendix A). This suggests that *F. luteovirens* QHU-1 is subject to strong purifying selection and relatively greater evolutionary pressure. While the distribution patterns of Ka/Ks values differ, the main peaks of *F. luteovirens* QHU-1 and its two closely related species are consistent. All three species are primarily distributed in the 0–1 range, indicating that they are functionally conserved in terms of transposon evolution.

### 3.6. Search and Analysis of Genes (Clusters) Involved in Secondary Metabolites

*F. luteovirens* is a large, edible fungus found on the Qinghai–Tibet Plateau. It has significant medicinal and economic value. We used antiSMASH to search and analyze the secondary metabolite BGCs in the genome. The results predicted that *F. luteovirens* QHU-1 contains 15 clusters and 17 core genes. These include seven terpenoid synthase-encoding genes, four NRPS-like genes, two NI-siderophore and PKS-encoding genes, one RIPP-like gene, and PKS and NRPS hybrid synthase-encoding genes. The 15 clusters are distributed across 10 chromosomes, and the core genes are relatively dispersed. Six chromosomes contain only one core-encoding gene each, while the remaining four chromosomes with multiple core-encoding genes have the highest number on Chr6A. This chromosome primarily contains terpenoid-encoding genes, while the two core-encoding genes on Chr2A are both NRPS-like types.

Given the critical role of core genes in secondary metabolite biosynthesis, we performed an in-depth analysis of the 17 core genes predicted from *F. luteovirens* QHU-1. First, BLAST alignment was performed on the seven predicted terpenoid synthesis-related genes, and six sesquiterpene synthases were selected. The sequences of these six sesquiterpene synthases were then compared with 58 known Agaricales order STSs using the maximum likelihood method to construct a phylogenetic tree. This tree grouped the six sesquiterpene synthases into four clusters (Figure 6B). Two enzymes were identified for each of the cyclization features: 1,10-cyclization of (2*E*,6*E*)-FPP and 1,6-cyclization of (3*R*/*S*)-NPP. The enzymes with 1,11-cyclization of (2*E*,6*E*)-FPP and 1,10-cyclization of (3*R*)-NPP each had one corresponding gene (1001698.1 and 1003064.1, respectively; Figure 6B). This finding suggests the diversity of sesquiterpene types in the *F. luteovirens* QHU-1. Based on the prediction results from antiSMASH, two PKS-encoding genes were identified in the genome of *F. luteovirens* QHU-1. An evolutionary tree was constructed using the 12 previously reported PKS-encoding genes from basidiomycetes, and the two genes were classified into two distinct categories based on their catalytic substrates (Figure 6C). The gene encoded by 1002903.1 is capable of forming anthraquinone compounds. Multi-domain analysis revealed that this enzyme contains five domains (KS-AT-PT-ACP-TE) (Figure 6D). Multi-domain analysis identified 100383.1 as a PKS-NRPS hybrid enzyme containing four domains (KS-AT-PT-A), showing significant differences in domain composition compared to other PKS enzymes in the same branch of the phylogenetic tree (Figure 6C,D). This suggests that the substrates catalyzed by this enzyme may exhibit greater structural diversity than those of other enzymes.

## 4. Discussion

As a global biodiversity hotspot, Qinghai Province, with its complex topography (encompassing plateaus, mountains, and hills and a wide range of altitudes) and unique climatic conditions (low temperatures and strong ultraviolet radiation), has given rise to highly specialized fungal communities and diverse microbial ecological environments [31]. Our research team previously focused on the lichen symbiotic microorganisms around Qinghai Lake, from which 27 actinomycetes with distinct morphological characteristics were isolated, and their potential antibacterial activities and biosynthetic potentials were assessed [32]. Subsequent research further led to the isolation of a *Streptomyces* strain QHH-9511, with potential anti-MRSA activity from Qinghai lichen symbionts [33]. Recently, the first systematic survey of macrofungi resources in Qinghai Province has made significant progress, with a total of 807 macrofungi species identified, including a large number of fungi with high edible and medicinal value as well as potential toxicity [30]. The survey results indicate that the Qilian Mountains and the Three Rivers Source area are the main regions of macrofungal diversity in Qinghai Province [34,35]. However, despite the macrofungal resource survey filling an important gap, the traditional focus of microbial research in Qinghai Province has still been relatively concentrated on actinomycetes [36], *Streptomyces* [37], and *Bacillus* [38,39,40], with research on the resource exploration, ecological functions, and development and utilization of macrofungi (except for some entomopathogenic fungi [41,42]) still being insufficient.

As an important biological resource and local specialty on the Tibetan Plateau, *F. luteovirens* has been used to treat swelling since ancient times and holds a high status in the Tibetan medical system. It is also highly popular among the local people due to its delicious taste and high nutritional value, and the evaluation of its bioactivity and nutritional value is one of the current research hotspots. Although two genomes of *F. luteovirens* have been reported so far, their assembly quality is not satisfactory. This study for the first time used the HI-C technology to complete the high-precision assembly of the *F. luteovirens* genome, successfully assembling the genome to the chromosomal level. The two obtained genomes were 26.77 Mb in size with an N50 length of 23.44 Mb and 27.04 Mb in size with an N50 length of 23.23 Mb, respectively. The BUSCO completeness reached 97.6%, indicating that the genome assembly quality is good and significantly better than the other two reported genomes [10,11]. The LTR insertion time and Ka/Ks analysis results show that *F. luteovirens* QHU-1 has no obvious continuous insertion in the past 30 million years and is undergoing relatively strong purifying selection, with a relatively greater evolutionary pressure. This study has to some extent solved the dilemma of the lack of a high-quality genome in *F. luteovirens* and laid a solid foundation for the subsequent exploration of active secondary metabolites and the innovative development of *F. luteovirens*.

Initially, *F. luteovirens* was classified within the genus *Armillaria*. Subsequently, through the construction of a phylogenetic tree using 16S rRNA, it was reclassified into the genus *Phlebopus*. However, no one has systematically analyzed the differences between *F. luteovirens* and *Armillaria* species. In this study, we conducted phylogenetic and homologous protein comparison analyses between *F. luteovirens* and ten *Armillaria* species, revealing the distinctions between *F. luteovirens* and *Armillaria*. A considerable number of recent studies have found that the secondary metabolites of *F. luteovirens* possess various bioactivities, including antioxidant and antitumor properties. We carried out relevant analyses on the BGCs of its secondary metabolites and performed phylogenetic analyses on the functions and classifications of its sesquiterpene and polyketide synthases, inferring their potential functions and completing the evaluation of the biosynthetic potential of *F. luteovirens* secondary metabolites.

The chromosome-level genome assembly of *F. luteovirens* presented here demonstrates the potential of advanced genomics to transform our understanding of macrofungal biology, particularly for understudied medicinal species. Historically, the presence of taxonomic ambiguity in groups such as *Armillaria*-allied fungi has impeded the precise estimation of evolutionary relationships. Our phylogenomic analysis leveraging 78 conserved orthologs across 32 basidiomycetes (Figure 4) resolved the 90.6 MYA divergence of *F. luteovirens* from *Armillaria* and established its sister relationship with *Mycocalia*/*Crucibulum*, a finding inaccessible through traditional markers like ITS [30]. The availability of such high-resolution phylogenies is contingent on contiguous assemblies, as fragmented genomes impede accurate ortholog detection and synteny-based comparisons (e.g., synteny disruption in prior assemblies [10,11] versus our Hi-C-anchored chromosomes). It is imperative to acknowledge that for medicinal fungi, genome sequencing serves to unravel the “black box” of bioactive compound biosynthesis. The identification of 15 BGCs (Table 2) has provided a genetic blueprint for the antioxidant and antitumor activities of *F. luteovirens*, including phylogenetically divergent sesquiterpene synthases (see Figure 5B) and a structurally innovative PKS-NRPS hybrid (Figure 6D). Comparable advances are being achieved in other macrofungi of pharmacological significance. For example, the genomes of *Ganoderma* species have been found to possess mechanisms that contribute to the diversification of terpenoids [43,44]. Furthermore, research has indicated that the genome of *Cordyceps sinensis* comprises BGCs linked to its biological and pharmacological effectiveness [45,46]. In contrast, in *F. luteovirens*, the terpene synthase clade that specializes in 1,10-cyclisation of (2*E*,6*E*)-FPP (1001698.1, Figure 6B) may underlie its anti-inflammatory sesquiterpenoids. This hypothesis can now be tested via heterologous expression guided by domain annotations. It is evident that, in addition to resolving evolutionary histories, chromosomal resources function as a foundation for the identification of novel pathways, the engineering of strains, and the sustainable production of fungal therapeutics.

## 5. Conclusions

This study, for the first time, utilized the HI-C technology to achieve high-quality assembly and haplotype orientation of the *F. luteovirens* genome from the Qilian Mountains in Qinghai Province, successfully assembling it onto 13 chromosomes. Phylogenetic analysis indicates a substantial divergence in differentiation time from the genus *Armillaria*, with the closest relatives being *Mycocalia denudata* and *Crucibulum laeve*, from which it diverged approximately 90.6 MYAs. Gene family expansion and contraction analysis reveals significant changes in gene families during their evolution. However, LTR insertion timing suggests that its transposons are more stable than those of the other two strains. This stability may be attributed to the influence of the strong ultraviolet radiation and low temperatures of the Tibetan Plateau, which likely result in lower activity of transposase enzymes. On the other hand, in this study, we conducted relevant analyses on the BGCs of secondary metabolites in *F. luteovirens* QHU-1, predicting 15 BGCs, including 17 core genes such as terpene synthases, PKSs, and NRPSs. We performed phylogeny-based clustering analysis on sesquiterpene synthases and polyketide synthases to predict their functions and predicted the domains of PKS, thereby laying the foundation for the subsequent evaluation of the biosynthetic potential of *F. luteovirens*.

## Figures and Tables

**Figure 1 jof-11-00621-f001:**
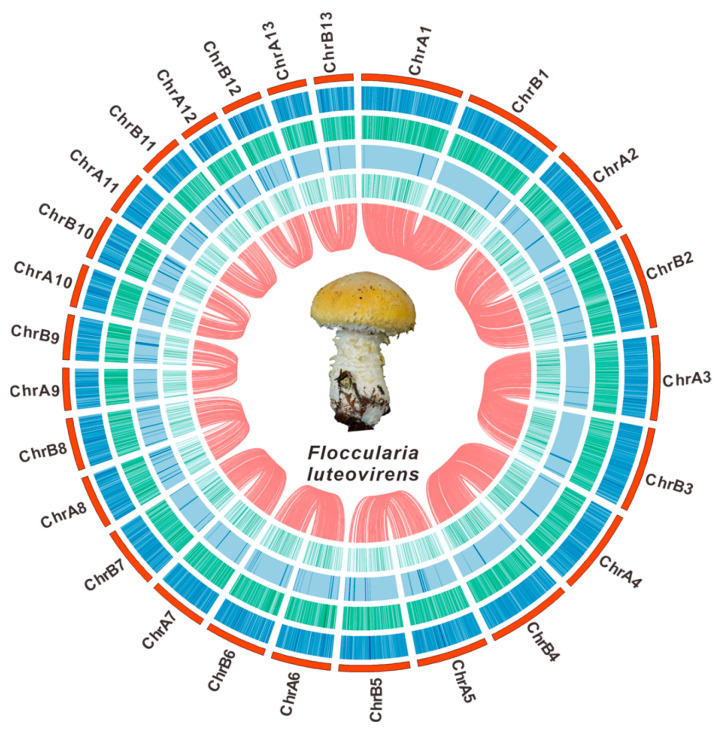
The genomic characteristics of *F. luteovirens* QHU-1. From outer to inner: I. chromosomes; II–IV. GC density, GC skew, and AT skew (window size 1 kb); V. gene density (window size 1 kb). The red lines inside indicate the relationships between the corresponding genomes, and the central circle shows a photograph of the fruiting body of *F. luteovirens*.

**Figure 2 jof-11-00621-f002:**
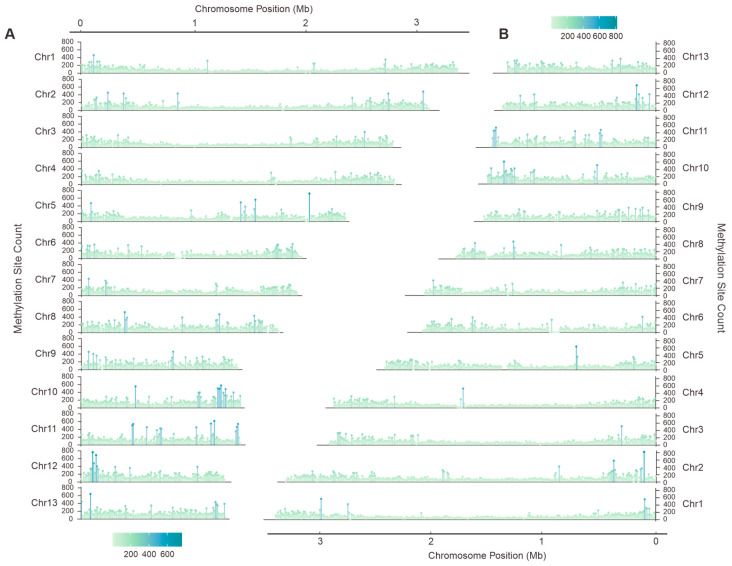
Single nucleotide polymorphism (SNP) analysis of *F. luteovirens* QHU-1. (**A**,**B**) are haplotype A and haplotype B, respectively.

**Figure 3 jof-11-00621-f003:**
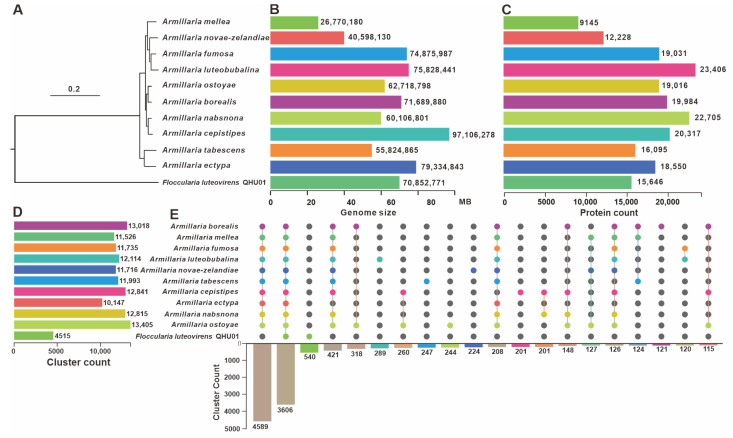
Comparative genomic analysis of *F. luteovirens* QHU-1 and ten other fungi of the genus *Armillaria*. (**A**) Phylogenetic analysis. (**B**) Genome size statistical analysis. (**C**) Protein count statistical analysis. (**D**) Cluster count statistical analysis. (**E**) Homologous protein comparison analysis. Colored dots correspond to species that contain this type of gene. Lines indicate that multiple species share this gene. The bar chart below shows the amount of homologous proteins present.

**Figure 4 jof-11-00621-f004:**
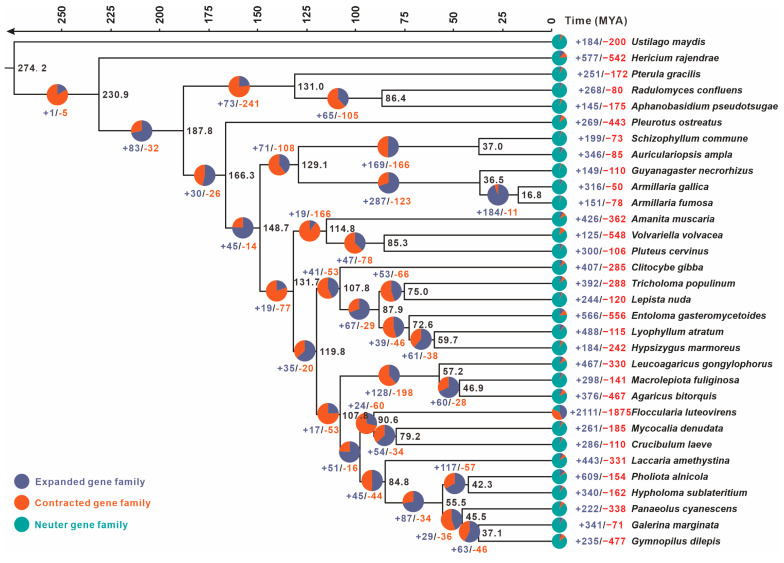
The evolutionary relationships and expanding and contracting gene families of *F. luteovirens* QHU-1 and the remaining 31 representative basidiomycetes were investigated. A maximum likelihood credibility tree was inferred from 78 single-copy orthologous genes. All nodes were supported by sufficient evidence. Divergence times were annotated as average crown ages for each node. Black numbers on branches indicate corresponding divergence times (MYA).

**Figure 5 jof-11-00621-f005:**
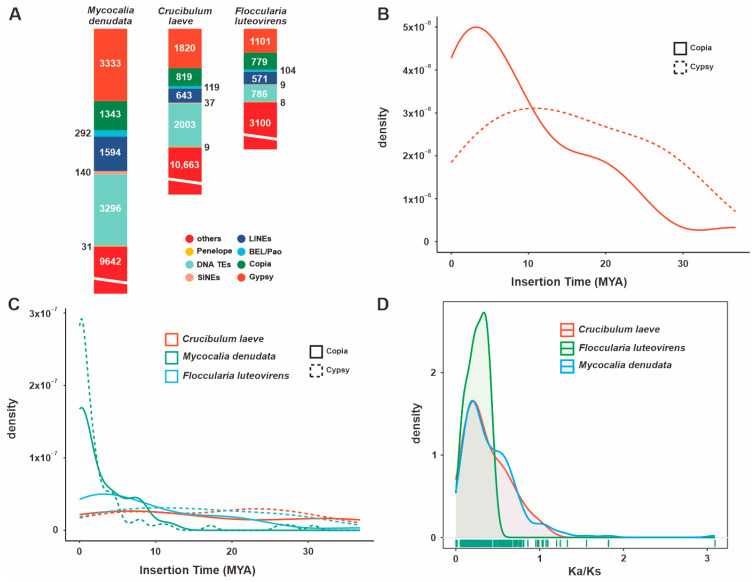
Analysis of TEs and positively selected genes in the *F. luteovirens* QHU-1 genomes and two closely related taxa. (**A**) A comparison of TE families in their taxa. (**B**) Insertion bursts of Gypsy and Copia elements in *F. luteovirens* QHU-1. (**C**) Comparison of temporal patterns of intact LTR-RT insertion bursts in their taxa. (**D**) The frequency distributions of Ka/Ks are shown between homologous gene pairs of the three taxa.

**Figure 6 jof-11-00621-f006:**
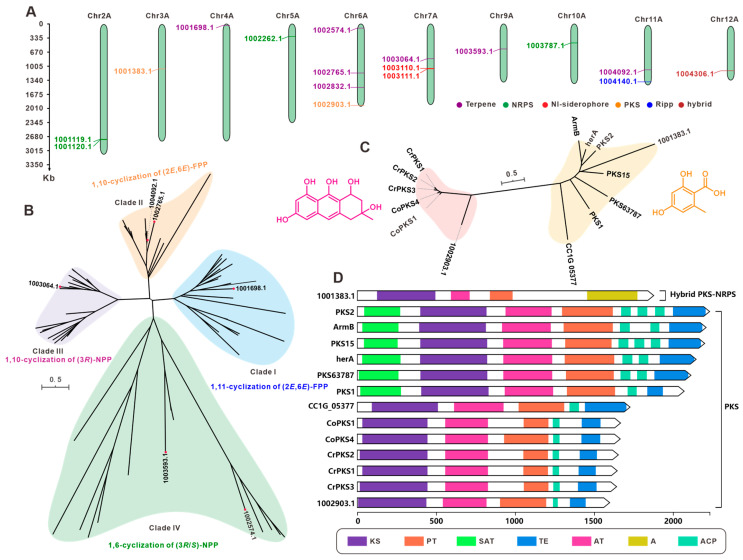
The core genes involved in secondary metabolite biosynthesis from *F. luteovirens* QHU-1. (**A**) Distribution of biosynthetic core genes for natural products on the chromosomes. (**B**) Phylogenetic tree analysis for STSs. (**C**) Phylogenetic tree analysis for PKS. (**D**) Domain characterization of the core enzymes containing multiple domains.

**Table 1 jof-11-00621-t001:** Genomic comparison within *Floccularia* species.

Subspecies Number	QHU-1	NWIPB-YM1807	FLZJUC10
Sequencing technology	PacBio, Illumina	PacBio, Illumina	PacBio, Illumina
Sequencing depth		171.93×	190.0×
No. of contig	NA	183	23
No. of chromosome	14	NA	NA
Total length (bp)	26,770,180	28,778,388	27,003,024
Largest length (bp)	3,345,222	2,429,293	3,288,420
Contig N50 (bp)	2,344,500	571,000	2,275,160
BUSCO completeness (%)	97.6	93.9	89.3
GC content (%)	43.54	43.36	43.5
No. of protein-coding genes	4545	8333	7068
GenBank accession No.	PRJNA1268684	GCA_004012055.1	GCA_009739215.1
Reference	This study	Gan et al. [10]	Liu et al. [11]

NA indicates not available.

**Table 2 jof-11-00621-t002:** Putative BGCs responsible for secondary metabolites in haplotype A of the *F. luteovirens* QHU-1 genome.

Cluster No.	Location	Start (bp)	End (bp)	Core Gene ID	Core Gene Type
1	Chr2A	2,639,131	2,764,041	1001119.1 1001120.1	NRPS-like
2	Chr3A	1,036,214	1,145,824	1001383.1	PKS
3	Chr4A	1719	111,756	1001698.1	terpene
4	Chr5A	263,686	362,057	1002262.1	NRPS-like
5	Chr6A	79,319	130,487	1002574.1	terpene
6	Chr6A	1,144,716	1,176,558	1002765.1	terpene
7	Chr6A	1,486,491	1,514,860	1002832.1	terpene
8	Chr6A	1,899,609	1,954,623	1002903.1	PKS
9	Chr7A	775,776	847,036	1003064.1	terpene
10	Chr7A	1,035,633	1,077,322	1003110.1 1003111.1	NI-siderophore
11	Chr9A	588,607	610,148	1003593.1	terpene
12	Chr10A	433,137	477,682	1003787.1	NRPS-like
13	Chr11A	1,039,901	1,062,249	1004092.1	terpene
14	Chr11A	1,312,927	1,375,455	1004140.1	RiPP-like
15	Chr12A	1,018,312	1,078,002	1004306.1	NRPS-like PKS

## Data Availability

The original contributions presented in this study are included in the article/Appendix A. Further inquiries can be directed to the corresponding authors.

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
