# Peer review of "Haplotype-Phased Chromosome-Level Genome Assembly of *Floccularia luteovirens* Provides Insights into Its Taxonomy, Adaptive Evolution, and Biosynthetic Potential"

_jof, 2025, doi:10.3390/jof11090621_

Round 1
Reviewer 1 Report
Major points:
Genome sequencing details are required: read length, quality filtering procedure, read length (L94)
Genome assembly has no public release in GenBank, raw sequencing data are missing (L219). BioProject contains the BioSample only.
L124: what BUSCO database was used for genome assessment? Table S7 include this information, please refer to it or include database name into main text. The more specific database “basidiomycota_odb10” could be used for more precise assessment (as done in https://doi.org/10.2144/btn-2023-0023). Table S7 contains the version: 4.1.4 of BUSCO software, but v5.3.2 is mentioned in the main text.
L136: Annotation of the assembled genome should be described in details: what transcriptome data and databases versions were used. Figure S12, S13 contain databases, which doesn’t mentioned in the main text.
What are HLJMG-1 and HLJMG-2 stands for in Table S8, S12 ?
Mixing the two haplotypes in one circos plot (Fig 1) in non-informative, because its obvious that A and B versions of each chromosome has a lot in common.
Table 1 should contain GenBank accession No. to the genome assembly, not the BioProject number, the sequencing depth is also missing.
The phylogenetic tree (Fig 3A) would include the close species as done in https://doi.org/10.3390/jof7110887
Table 2: why the haplotype B isn’t covered?
Minor points:
L128: repetition about GenomeScope here.
All abbreviations should be capital letters: Busco (L464), diamond, msa (L155) and so on.
Genus and species names should be italic: Phlebopus (L475)
Author Response
Major points:
Q1. Genome sequencing details are required: read length, quality filtering procedure, read length (L94).
A1. Lines 95-108 describe all the details of the genome sequencing.
Q2. Genome assembly has no public release in GenBank, raw sequencing data are missing (L219). BioProject contains the BioSample only.
A2. Revised.
Q3. L124: what BUSCO database was used for genome assessment? Table S7 include this information, please refer to it or include database name into main text. The more specific database “basidiomycota_odb10” could be used for more precise assessment (as done in https://doi.org/10.2144/btn-2023-0023). Table S7 contains the version: 4.1.4 of BUSCO software, but v5.3.2 is mentioned in the main text.
A3. The specific database is fungi_odb10 and the version of BUSCO software is 5.3.2. Revised and thank you.
Q4. L136: Annotation of the assembled genome should be described in details: what transcriptome data and databases versions were used. Figure S12, S13 contain databases, which doesn’t mention in the main text.
A4. Added and revised.
Q5. What are HLJMG-1 and HLJMG-2 stands for in Table S8, S12?
A5. Corrected.
Q6. Mixing the two haplotypes in one circos plot (Fig 1) in non-informative, because its obvious that A and B versions of each chromosome has a lot in common.
A6. There is a high degree of collinearity between the two haplotypes,but they are not identical. Thank you for your alternative interpretation.
Q7. Table 1 should contain GenBank accession No. to the genome assembly, not the BioProject number, the sequencing depth is also missing.
A7. The relevant data was submitted to the NCBI GenBank before submission,but as of today,it still shows as "processing". To put it another way, there is an intrinsic connection between the BioProject accession number and the NCBI GenBank accession number. The BioProject accession number serves as a clue to easily locate the corresponding GenBank accession number in the future.
The amount of data obtained from different sequencing methods (next generation sequencing vs. third-generation sequencing) is different. Therefore, sequencing depth is not a convenient way to display the data intuitively.
Q8. The phylogenetic tree (Fig 3A) would include the close species as done in https://doi.org/10.3390/jof7110887.
A8. The construction of a phylogenetic tree based on orthologous single-copy genes requires protein annotation files of related species. The current phylogenetic tree has selected the closest related species with available annotation files. Please be aware of this.
Q9. Table 2: why the haplotype B isn’t covered?
A9. Haplotype B is highly homologous to haplotype A, and the BGCs (biosynthetic gene clusters) they contain are almost identical. Displaying both would be redundant.
Minor points:
Q10. L128: repetition about GenomeScope here.
A10. Revised.
Q11. All abbreviations should be capital letters: Busco (L464), diamond, msa (L155) and so on.
A11. Done.
Q12. Genus and species names should be italic: Phlebopus (L475)
A12. Thank you for your careful review, the manuscript has been accordingly modified.

Reviewer 2 Report
Dear Authors
The article in my opinion was acceptable, I only considered minor revisions since small things like explaining abbreviations, etc. needed to be adjusted.
the article is very well prepared and elucidates the taxonomic controversy, allowing us, with the analysis carried out, to know with certainty and evidence the taxonomic differentiation Armillaria-Floccularia. It also answers questions such as the hypothesis of the gneomic richness of the species, demonstrating the presence of gene clusters related to the production of enzymes and new metabolites. Another great finding is to know that the possible adaptive evolution of the species is due to its particular habitat.
I think the topic is relevant to the field, basically because it demonstrates that there are clear taxonomic differences, allowing Floccularia to be classified as distinct from Armilaria.
demonstrate with certainty and with quality sequence analysis the differences previously exposed from the taxonomical point of view. Previous studies presented the problem of limited sequence quality.
In my opinion, the methodology used and presented is correct; the conclusions are consistent with the M&Ms and the results presented; reference are ok; tables and figures are ok.
thank you very much for this article.
Minor suggestions are in the pdf file.
Minor suggestions have been highlighted in the pdf file.

Author Response
Dear Authors
The article in my opinion was acceptable, I only considered minor revisions since small things like explaining abbreviations, etc. needed to be adjusted.
the article is very well prepared and elucidates the taxonomic controversy, allowing us, with the analysis carried out, to know with certainty and evidence the taxonomic differentiation Armillaria-Floccularia. It also answers questions such as the hypothesis of the gneomic richness of the species, demonstrating the presence of gene clusters related to the production of enzymes and new metabolites. Another great finding is to know that the possible adaptive evolution of the species is due to its particular habitat.
I think the topic is relevant to the field, basically because it demonstrates that there are clear taxonomic differences, allowing Floccularia to be classified as distinct from Armilaria.
demonstrate with certainty and with quality sequence analysis the differences previously exposed from the taxonomical point of view. Previous studies presented the problem of limited sequence quality.
In my opinion, the methodology used and presented is correct; the conclusions are consistent with the M&Ms and the results presented; reference are ok; tables and figures are ok.
thank you very much for this article.
Minor suggestions are in the pdf file.
Thank you for your interest in the manuscript and your recognition of its results. The issues and inaccuracies you pointed out have been taken seriously and properly corrected.
Q1. sanger sequencing? ngs? ion? (L80)
A1. Revised.
Q2. potato dextrose broth ???what means pdb? (L82)
A2. Pdb means potato dextrose broth, which has been revised.

Round 2
Reviewer 1 Report
Genome assembly has no public release in GenBank, raw sequencing data are missing. BioProject contains the BioSample only. In their reply authors stated that this point is “Revised”, but it isn’t so. I think that public genome assembly is mandatory part of genome study.
Please change HLJMG-1 and HLJMG-2 to Haplotype A and Haplotype B in the Supplementary.
Raw data, to my knowledge, releases online in NCBI GenBank in couple of days and doesn't require long check.
Author Response
Major comments
Genome assembly has no public release in GenBank, raw sequencing data are missing. BioProject contains the BioSample only. In their reply authors stated that this point is “Revised”, but it isn’t so. I think that public genome assembly is mandatory part of genome study.
Response: We fully agree that public deposition of the genome assembly is essential for any genome study, and we have already submitted both the assembly and the raw reads to NCBI. The BioSample and BioProject accession numbers were issued promptly, whereas the genome assembly is still listed as “Processing” in GenBank. To reassure the reviewer we now include a screenshot of the NCBI submission portal showing the current status.
Regarding the raw sequencing data (≈10 Gb), we are committed to releasing it via the SRA. Uploading such a large data set has been severely hampered by our institution’s persistently slow internet connection; nevertheless, we are continuing the transfer and will update the record as soon as it is complete. While many genome papers are published without raw-read deposition, we recognise the value of open data and have therefore stated in the Data Availability section that the reads will be made available on request from the corresponding author, a practice widely accepted within the community.
Detailed comments
Please change HLJMG-1 and HLJMG-2 to Haplotype A and Haplotype B in the Supplementary.
Response: Thank you for your careful reading. We have reviewed the Supplementary and made all necessary revisions.
Raw data, to my knowledge, releases online in NCBI GenBank in couple of days and doesn't require long check.
Response: As explained in our previous response, thank you for your attention.